# Review of Effects of Dam Construction on the Ecosystems of River Estuary and Nearby Marine Areas

**Xuan Zhang [1], Changling Fang [1], Yuan Wang [1], Xiaoyi Lou [1], Ying Su [2,\*] and Dongmei Huang [1,\*]**

[1] Ministry of Agriculture and Rural Affairs, East China Sea Fisheries Research Institute, Chinese Academy of Fishery Sciences, Shanghai 200090, China; zhangxuan@ecsf.ac.cn (X.Z.); fangling034081@163.com (C.F.); wangyuan8@163.com (Y.W.); huoxingmayi@126.com (X.L.)

[2] College of Ocean Science and Technology, Dalian University of Technology, Dalian 116024, China

\* Correspondence: yingsu@dlut.edu.cn (Y.S.); hdm2001@126.com (D.H.)

**Abstract:** Dams have made great contributions to human society, facilitating flood control, power generation, shipping, agriculture, and industry. However, the construction of dams greatly impacts downstream ecological environments and nearby marine areas. The present manuscript presents a comprehensive review of the influence of human activities on the environment, especially the effect of dam construction on the ecosystems of river estuaries and nearby marine areas, so as to provide a scientific basis for ecological environment protection. To summarize these impacts, this review used recent studies to comprehensively analyze how dam construction has affected river hydrology, geomorphology, and downstream ecosystems globally. Effects of dams on ecosystems occur through reduced river flow, reduced sediment flux, altered water temperature, changed estuary delta, altered composition and distribution of nutrients, altered structure and distribution of phytoplankton populations, habitat fragmentation, and blocked migration routes in river sections and adjacent seas. Effects of dam construction (especially the Three Gorges Dam) on the Yangtze River were also reviewed. Performing community and mitigation planning before dam construction, exploring new reservoir management strategies (including targeted control of dam storage and flushing sediment operations), banning fishing activities, and removing unnecessary dams (obsolete or small dams) are becoming crucial tools for ecosystem restoration.

**Keywords:** dam; river estuary; ecosystem; effect

## 1. Introduction

Dam construction has a long history, especially in China, where dams have been utilized since 3000 BC. People build dams mainly for river control, flood control, irrigation, hydropower, and shipping. According to statistics from the International Commission on Large Dams, as of April 2020, there were 68,000 large dams with heights of over 15 m or impounding more than 3 million m³ in the world (38,000 of which were located in China, accounting for 56% of the world's dams) (Figure 1) [1,2]. There were 53,544 dams in the range of 15–30 m high (31,666 in China), accounting for 78.7% of the global dams (46.6% in China). There were 77 dams over 200 m high (20 in China) around the world, accounting for 1.13% of the total dams. The total storage capacity of these dams approached 8000 km³, which is equivalent to 10% of the annual runoff of the world's rivers. So far, 50% of the world's rivers have been controlled or altered by hydraulic projects before reaching the ocean [3]. The Yangtze River is the largest river in China and the third-largest in the world. It winds 6300 km through the $1.8 \times 10^6$ km² Yangtze River Basin before reaching the East China Sea [4]. The Yangtze River Basin is an important economic zone in China with an abundance of resources, large population, and developed economy [5,6]. Therefore, the development and management of the Yangtze River are of great importance to the promotion and development of China's economy. The Yangtze River estuary and its adjacent sea area, where brackish and freshwater intensely mix, have a unique environmental structure and

serve a variety of functions [7,8]. There are 1811 large and medium-sized dam reservoirs on the Yangtze River. These dams have had a significant impact on the environment, but extensive and in-depth research is required to accurately characterize these effects [9–11]. At present, there are 50,000 dam reservoirs within the Yangtze River Basin, accounting for more than half of all the reservoirs in China [12]. The total water storage has reached 140 km$^3$, which accounts for 15.6% of the annual runoff of the Yangtze River [13,14]. Dam construction has altered the runoff and sediment load of the Yangtze River, changed the flux and composition of nutrients, and affected ecological systems throughout the Yangtze River Basin. The impacts of dam construction are not limited to its direct effects on downstream hydromorphology, ecosystems, and human life, but extend to the ecological environments of the estuary and its adjacent marine area [15,16].

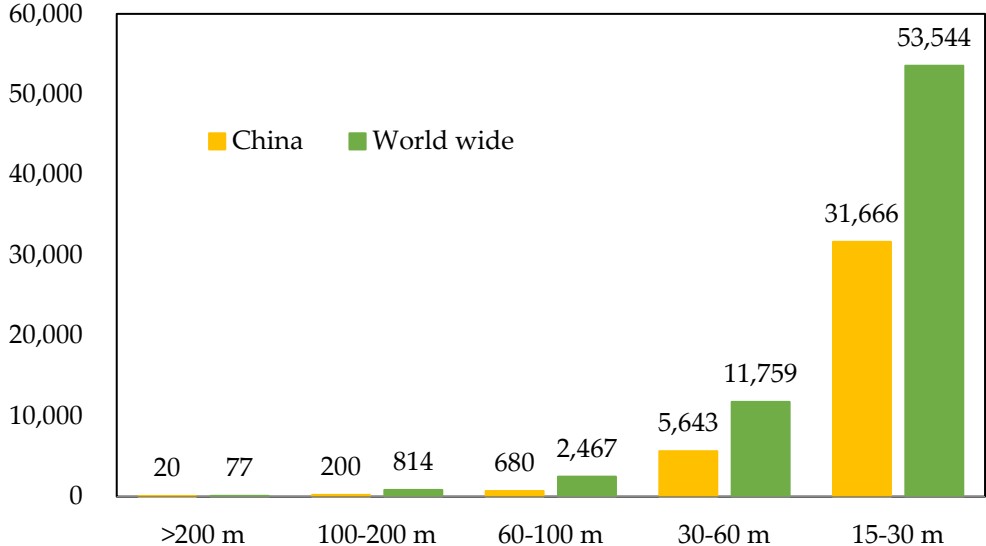

**Figure 1.** Number of registered dams worldwide and China.

Dam construction is a large part of human engineering infrastructure, which is the foundation for much of our daily lives. Damming rivers provides numerous conveniences for human societies. Dams and reservoirs can store water during rainy seasons and later release it to provide a consistent discharge and maintain sufficient flow throughout the year [2]. However, while developing natural resources to provide better living conditions for humankind, we must also ensure that the associated environmental impacts are monitored and limited.

As a part of integrated watershed management, dam construction presents both opportunities and challenges. Dam closures convert dynamic rivers into static reservoirs, which affects the hydrography and morphological evolution of rivers by altering flow velocity, water quality, temperature, turbidity, particulate matter, and other physicochemical parameters of rivers [2,17,18]. Dams also result in major anthropogenic disturbances to the biogeochemical cycles of nutrients which affect downstream wetlands, estuaries, underwater deltas, and adjacent marine ecosystems [3,19]. In addition, the recent global warming, in terms of temperature and precipitation, may exacerbate negative effects on the ecological environment by dam construction [20,21]. In 1997, the International Commission on Large Dams published a document, 'Position Paper on Dams and the Environment', which pointed out that improving environmental awareness was one of the most important developments at the end of the 20th century [1]. In 2016, the Chinese government put forward the concept of "Great Protection of the Yangtze River", which made the ecological restoration of the Yangtze River Basin a priority concept that now pervades all related work. How to balance the relationship between the ecological environment and dam construction has become the focus of people's attention [22,23].

Unfortunately, because of the long distance covered by the river before it meets the ocean, and various other anthropogenic disturbances within the river basin, the roles of dams in marine ecosystems can easily be ignored. Long-term observation and research are necessary to assess the impacts of dams on ecosystems of estuarine and adjacent sea areas [24–26]. This paper provides a review of recent studies on river damming and its impacts on ecosystems of estuarine and adjacent sea areas. This review aims to: (1) discuss changes in rivers by comparing periods before and after dam construction and relate these changes with sediment dynamics, nutrient fluxes, and estuarine ecosystems where rivers meet the sea; (2) identify the future directions and priorities for research in this field; and (3) provide a scientific basis for the protection of the ecological environment.

## 2. Methodology

Due to the high-quality and comprehensive records, the systematic bibliometric analyses were performed by the ISI Web of Knowledge database [27,28]. A preliminary search was performed to collect articles related to the research topic. The purpose of this search was to establish a framework for subsequent qualitative and quantitative analysis. The search string related to the study topic included "dam", "effect", "impact", "ecosystem", "estuary", "marine", and "sea". References in relevant articles were then tracked to check for the literature not detected in the Web search progress. The subject areas were Agricultural and Biological Sciences, Environmental Science, and Earth and Planetary Sciences. All retrieved files of journal papers, books, newspapers, and conference articles (published between 2000 and 2021) were further evaluated and filtered by searching the results to make sure: (1) they all related to the study topic; (2) all the dam effects focused on the estuary and nearby marine areas; (3) all the articles were published. Finally, all the related files were classified into six categories: flow flux, sediment, geomorphology, nutrient, phytoplankton, and fish. Not all 235 relevant studies retrieved were cited in this paper because of similar conclusions. The structured and systematic approach was performed in order to ensure the objectivity of conclusions [29,30]. In the end, future research directions are pointed out to provide a scientific basis for the protection of the ecological environment.

## 3. Results

For the research region, there are many studies on dam reservoirs and downstream rivers, but few studies on the estuary and its adjacent marine area. For the research object, most published studies focused on the effect of dam construction on flow flux and sediment transport (151), with some dealing with nutrients (52) and biodiversity (32) in the river downstream and adjacent sea areas. The effect of dam construction on the environment of the river estuary and its adjacent marine area was summarized with some examples. Effects of dams on the ecosystems included regulating river flow, reducing sediment flux, changing water temperature of reservoirs, erosion of estuary deltas, altering the composition and distribution of nutrients, changing the structure and distribution of phytoplankton, and habitat fragmentation in downstream rivers and their adjacent sea areas (Figure 2 and Table 1).

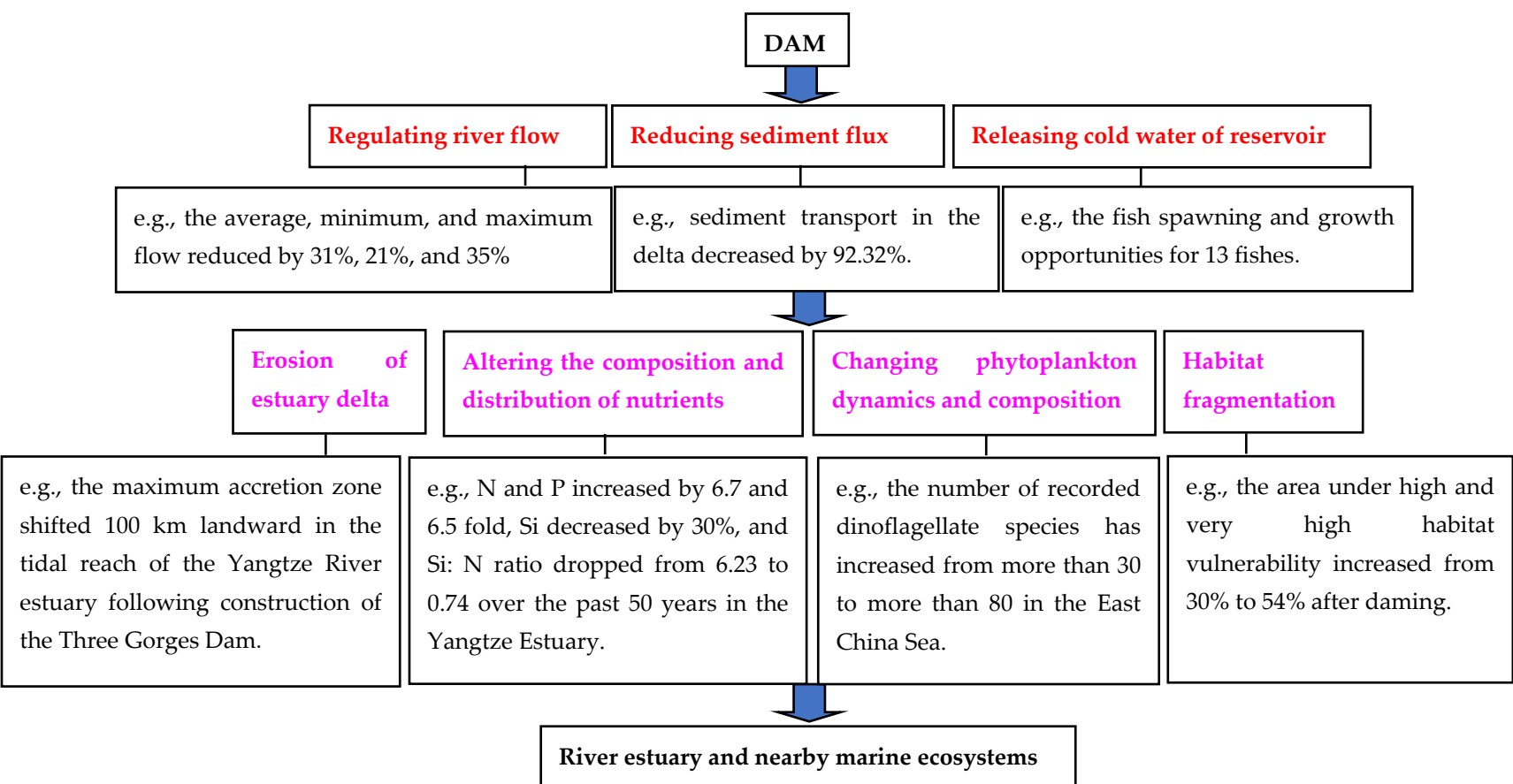

**Figure 2.** Effects of dam construction on river estuary and nearby marine area ecosystems [24,25,31–35].

**Table 1.** Studies about the effects of dam construction.

| Characteristics | Effects on the Downstream River and Coastal Water | Reference |
|---|---|---|
| Reduction of river flow | Reduced flow by 21–90% in downstream area. | [31,36,37] |
| Reduction of sediment transport | Reduced sediment transport by 75–92% in downstream area. | [32,38,39] |
| Changes in sediment dynamics and geomorphic processes | Retreatment of the estuarine turbidity maximum zone and erosion that degrades the underwater deltas in estuaries. | [26,40–43] |
| Nitrogen and phosphorus nutrient flux | Intercepted about 42–93% of river nutrients, especially phosphorus. | [44–47] |
| Silicon nutrient flux | Affected river flow velocity, less silicon supplemented from the continent, and the amount of silicon land–sea fluxes decreased by 50–80%. | [48–52] |
| Nutrient ratios | Reduced silicon:nitrogen ratios sharply from 6.23 to 0.74. | [25,53] |
| Phytoplankton community | Impacted the phytoplankton composition and increased the outbreak numbers of harmful red tides. | [54,55] |
| Plant | Decreased functional richness and species diversity in downstream areas. | [56–59] |
| Bird | Provided positive and negative influences on the habitat suitability for birds. | [60–63] |
| Fish | Caused habitat fragmentation and affected fish populations, species, and sizes. | [35,64,65] |

*3.1. Effects of Dam Construction on Sediment Flux into the Ocean*

Precipitation varies in different months under the influence of monsoon [66]. In recent years, frequent shifts in climate affect precipitation, which has a negative impact on agricultural production [67,68]. Dams could store water during wet seasons and release freshwater during dry seasons, stabilizing the water supply to support agricultural irrigation in delta areas [36]. A dam segments the river, changing it from a flowing whole to one impeded by a static reservoir that reduces river flow. The average, minimum, and maximum flow reduced by 31%, 21%, and 35% in the lower São Francisco Riverin after the construction of the Xingó reservoir [31]. After the construction of the Farakka dam in India, the river flow into the downstream Bengal Bay showed a deficit of 75% [37]. Similarly, the Aswan dam intercepted 90% of the Nile's runoff upon its construction, leading to the collapse of fisheries in the Mediterranean [69]. Besides, it has been reported that the dam operation affects the size and frequency of flow alteration [70]. Dam construction could influence the hydrological regimes of rivers by reducing the peak flow and changing the flow periodicity [71].

It has been reported that, in addition to impacting river flows, dams have important impacts on sediment dynamics and geomorphic processes [30,72–74]. Rivers are estimated to transport about 90% of all dissolved and particulate matter that is deposited in the ocean, and the total annual sediment transported by rivers is estimated at about 19 billion tons, representing a very important driver of hydrological changes and the morphological evolution of estuaries and adjacent sea areas [75,76]. Amenuvor et al. [32] studied the hydrology of the Volta River before and after the Akosombo Dam over the period from 1936 to 2018 using Landsat remote sensing images, and the results indicated that the establishment of the dam resulted in significant hydrological changes and altered the morphological evolution of the river. Their study showed that sediment transport and river flow in the delta decreased by 92.32% and 23.23%, respectively. In major rivers of China, it has been found that the annual total sediment transport to coastal areas has decreased from 2.03 billion tons in 1955–1968 to 0.50 billion tons in 1997–2010 [38]. Similarly, studies of sediment transport in Swiss rivers [39], Russian rivers [40], European rivers [41], North African rivers [42], and South East Asian rivers [43] have shown that anthropogenic disturbances in river basins (mainly dam and reservoir construction) are the main causes of reductions in sediment transport.

Dam construction changes the topography of the riverbank, increases the erosion of the downstream riverbed, and causes erosion that degrades underwater deltas in estuaries [26,77,78]. By altering sediment transport, dams can affect the benthic environments of estuarine areas, causing retreatment of the estuarine turbidity maximum zone. It has been reported that the estuarine turbidity maximum zone in Portugal moved 8–16 km upstream compared to previous records after the construction of the Alqueva Dam [64]. More attention should be given to the effects of dam construction on geomorphic processes for its relationship with sediment flux into the ocean.

The reservoir dams in the Yangtze River Basin have had little effect on the runoff of the Yangtze River. From 2002 to 2016, before and after the construction of the Three Gorges Dam, the annual runoff at the Yichang, Shashi, and Datong monitoring stations located in the downstream reaches of the Yangtze River increased by 5.28–8.55% [74]. However, as a consequence of the water storage function of the Three Gorges Dam, the monthly runoff of the Yangtze River has changed greatly due to seasonal regulation [79,80]. It is reported that about 77% of sediment was trapped by the Three Gorges Reservoir [76]. Dam construction has also significantly reduced sediment flux from the Yangtze River into the sea [81]. According to the long-term monitoring results of Datong Station, the annual runoff of the Yangtze River has not decreased, but instead, slightly increased. It was found that the sediment flux at Datong Station on the Yangtze River had three notable step-down reductions: one was in 1969 (the Danjiangkou Dam finished) when the sediment flux decreased from 490 million tons to 440 million tons; the second was in 1986 (a series of dams finished) when the sediment flux dropped to 340 million tons; and the third was in

2003 (Three Gorges Dam began operation) when the sediment flux decreased to 180 million tons. Overall, the sediment flux at Datong Station on the Yangtze River declined by more than 60% between 1968 and 2003 [82].

Most studies have focused on the effects of dam construction on sediment transport from the Yangtze River to the sea. Based on the results of field surveys before and after the closure of the Three Gorges Dam, the suspended sediment front zone in the Yangtze River estuary shifted to the low salinity direction after the Three Gorges Dam finished. Decreases in sediment flux will have an effect on the geomorphological characteristics of the downstream Yangtze River and its adjacent marine area [5,83]. Altered sediment fluxes can influence the evolution of downstream riverbeds, and shifts from sedimentation to erosion can lead to the degradation of coastal beaches and the erosion of underwater deltas [84]. The underwater delta of the Yangtze River estuary will not be eroded, only when the annual sediment flux of Datong station is greater than 270 million tons. However, the sediment flux of Datong station in 2006, 2007, and 2010 was less than this value. Therefore, with the increasing erosion, more sediment may be deposited in the estuary area rather than transported to the adjacent waters of the Yangtze River estuary. The deposition rates of underwater deltas in and out of the Yangtze River estuary decreased from 25 and 51 mm/yr in 1958–1978 to 14 and 2 mm/yr in 1978–1997, respectively [85]. Under the influence of the Three Gorges Dam, the underwater delta of the Yangtze River estuary will continue to be eroded and is expected to reach equilibrium around 2060 [86]. In addition, the maximum accretion zone shifted 100 km landward in the tidal reach of the Yangtze River estuary following the construction of the Three Gorges Dam due to the decreasing sediment flux [33].

### 3.2. Effects of Dam Construction on Nutrient Flux to the Sea

Artificial lakes formed by dam construction will affect the biogeochemical cycles of nutrients (carbon, nitrogen, phosphorus, silicon, etc.) in water [46,53]. The nutrients in the upstream reaches of a river are intercepted by the phytoplankton that flourish in the reservoirs [19]. This removes nutrients from the water and has far-reaching ecological impacts on the global biogeosphere. The effects of dams on riverine nutrient fluxes vary from one nutrient to another. About 42–93% of river nutrients can be intercepted by reservoirs [44], especially phosphorus (P), whose uptake in reservoirs ranges between 16% and 98% [45]. After the construction of the Three Gorges Dam, eutrophication in downstream reaches has been alleviated [87]. Although the dam intercepted some nutrients, the nitrogen (N) inputs to coastal waters could increase by 20% and with a doubling of P inputs in Indonesia by 2050 due to anthropogenic sources such as domestic sewage, industrial wastewater, and agricultural fertilization [88].

Unlike other essential nutrients, silicon (Si) does not have downstream sources and is not resupplied to rivers after dam interception [48,49]. Due to the effect of anthropogenic perturbations (mainly dam construction, industrial wastewater, and use of fertilizers containing N and P nutrients), the amounts of N and P have increased by 6.7 and 6.5 times, respectively, while the amount of silicon has decreased by 30% [47,89]. In addition to nutrient retention, Si nutrient concentrations in downstream rivers and coastal marine areas can be reduced by other hydrological changes caused by dams [48,76]. Under the blocking of the dam, the flow velocity of the river decreases, which weakens the bank erosion and reduces the water-ground interaction. As a result, less Si is supplemented from the continent, and the amount of silicon in the river decreases [50]. Humborg et al. [51] found that the major cause of the reduction in land–sea Si fluxes was dams. The dissolved silicate (DSi) yield of moderately dammed rivers was only 50% of the practically undammed river. It has also been reported that 80% of Si in the ocean has been imported from rivers [52]. Therefore, the global modification of riverine Si flux directly affects the distribution of ocean basin Si concentrations, especially for coastal marine areas.

High anthropogenic N and P nutrient loads reduce the Si: P and Si: N nutrient ratios [53]. Clearly, nutrient composition patterns have been fundamentally changed by

dams as they alter the absolute nutrient fluxes and the nutrient ratios of riverine and coastal marine areas.

The Yangtze River transports large quantities of macronutrients from land into the East China Sea. The annual flux of N, P, and Si nutrients was $1.43 \times 10^6$ t, $2.35 \times 10^5$ t, and $3.17 \times 10^6$ t, respectively [90,91]. This accounted for 15%, 12%, and 1% of the N, P, and Si fluxes intercepted by the Three Gorges Dam [46]. However, the annual amount of N and P nutrient loading into the East China Sea increased by 15.3% after the Three Gorges Dam construction was completed [87]. In addition, the nutrient front displacement to the low salt direction after the construction of the Three Gorges Dam, and the N and P nutrient contents of the nutrient front, showed an increasing trend [79,90]. This indicated that anthropogenic activities showed a more significant impact on the N and P nutrients entering the sea than the Three Gorges Dam. It is generally believed that the decrease in Si content is related to the decreased runoff from the Yangtze River [53].

The changes in N, P, and Si nutrient amounts resulted in a sharp decrease in the ratio of Si: N in the Yangtze River estuary, dropping from 6.23 to 0.74 over the past 50 years [25,53]. Since the Three Gorges Dam started storing water, the amounts of N and P increased by 27% and 60%, while Si decreased by half and the Si: N ratio decreased from 3.25 to 0.98 [92]. Survey results of the 2002–2006 cruises in the Yangtze River estuary area showed that the P: N and Si: N nutrient ratios had seriously deviated from the Redfield ratio after the construction of the Three Gorges Dam, especially in spring and summer [93]. The N nutrient is in surplus in the Yangtze River estuary, while Si and P are relatively low. These findings clearly demonstrate that dams fundamentally alter water quality and nutrient supplies in rivers.

### 3.3. Effects of Dam Construction on Ecosystems of the Estuary and Adjacent Coastal Area

Estuaries are complex amalgams of various material systems, structural systems, functional systems, and energy systems [94]. They are ecological transition zones and represent some of the most intense and complex land–sea interactions [95]. Macronutrients are carried by rivers from land to estuaries, promoting the growth and reproduction of marine and saltwater tolerant organisms and maintaining the highly complex and variable ecosystems [96]. The abundance of organic and inorganic elements in estuarine areas makes them ideal for primary productivity, as demonstrated by the plumes of highly productive areas fronting estuaries worldwide [97]. As some of the highest productivity zones in the ocean, many famous fishing grounds are associated with estuaries, such as the Lvsi and Zhoushan fishing grounds in the Yangtze River estuary. However, estuaries and adjacent sea areas are frequently densely populated with developed agriculture and industry, which makes estuarine ecosystems highly impacted by human activities.

Downstream, estuarine, and adjacent marine ecosystems will all be affected by damming. Phytoplankton in the ocean absorb nutrients in a constant ratio known as the Redfield coefficient. The deviation of the nutrient ratio from the Redfield coefficient in seawater will affect the growth and composition of phytoplankton [98,99]. Decreases in silicon concentrations and increases in nitrogen and phosphorus concentrations have been linked to changes in the growth and species composition of phytoplankton communities, as well as increases in the frequency of harmful red tide outbreaks [54,55]. Phytoplankton are important components of aquatic food webs, so changes in their abundance and composition can have effects on benthic animals, fishes, plants, and birds [56,100]. Worse still, marine products contaminated with algal toxins can cause illness or even death in humans if they are mistakenly consumed [101].

Plant species richness was significantly impacted by dam construction [57,59,102]. Most studies in the research of dam effects focused on the plants in the reservoir, downstream channel, and lake. Moreover, studies about the effects of dams on plants in coastal waters were insufficient. The water level and plants could be impacted by dam construction, which would further affect the habitat suitability of birds [103–105]. A recent study claimed that the Three Gorges Dam artificially generated ecological water levels, which

provided benefits for birds [60]. Besides, the effects of dams on flow changes created stable habitat for American dippers [61]. However, some studies indicated that dam construction would have a negative influence on the habitat suitability for birds by destroying coastal wetlands [62,63].

Changes in freshwater flow by dams affect the estuary salinity, resulting in changes in salinity-sensitive marine organisms [106–108]. For instance, the low estuary salinity caused by the Fitzroy Barrage could reduce estuarine jellyfish populations because jellyfish cannot survive at low salinities [109]. Habitat fragmentation could be impacted by damming. The area under high habitat vulnerability increased from 30% to 54% after damming wetlands in the Tangan river basin of India [35]. Damming can decimate local fishing industries by reducing the abundance of important commercial fish [24,27]. For example, dam construction caused a shift in the phytoplankton in the Guadiana estuary of Portugal from diatoms to flagella and cyanobacteria. This shift caused a corresponding change in the species of fish, from those that eat diatom phytoplankton (e.g., anchovies) to those that eat non-diatom phytoplankton, and the abundance of anchovy eggs decreased by 99.99% [64]. A similar case occurred in the Brazilian Amazon, where fish production and diversity were all impacted by dam construction [65].

Macronutrients carried by the Yangtze River can lead to eutrophication in adjacent marine areas. After the construction of the Three Gorges Dam, the chlorophyll peak in the Yangtze River estuary shifted to the direction of low salinity. The altered flow fluxes, sediment fluxes, and nutrient supply patterns could affect the quantity and structure of phytoplankton, zooplankton, and even the benthic communities [110]. The dam construction has greatly impacted the distribution of chlorophyll *a* in the Yangtze River estuary [34]. Since the 1970s, large-scale red tides have occurred regularly in the marine waters near the Yangtze River estuary. The frequency of red tides has tripled every 10 years, with a total of 748 red tide disasters from 2000 to 2019, resulting in an increasingly large cumulative disaster area [111]. This increase in the frequency of toxic red tide outbreaks year by year has occurred due to the increasing anthropogenic N supply and decreasing Si concentrations in the Yangtze River estuary and its adjacent marine areas [112,113]. Since 2000, the number of recorded dinoflagellate species in phytoplankton communities has increased from more than 30 to more than 80. At present, *Prorocentrum* ranks as the most frequent and disaster-associated red tide species and has caused serious economic losses to fishing and aquaculture in the East China Sea [34]. Conversely, diatoms once accounted for 33% of the phytoplankton community, but decreased to 24% between 1980 and 2002 [114]. Red tide algae can have lethal effects on important commercial species like clams, shrimp, and mullets [115].

It has been reported that harmful algal toxins can be transferred and accumulated through food webs, thus harming fish, birds, and other marine organisms [116,117]. Furthermore, the Three Gorges Dam construction could cause fish biodiversity to decline. The temperature decreased by about 0.4 °C after the completion of the Three Gorges Dam, which changed the spawning time of the Chinese sturgeon [102].

According to the hydrology, biology, chemistry, and sedimentation in the Yangtze River estuary, the Yangtze diluted water can be divided into plume water (salinity of 5–25 and sediment of 100–500 mg/L) and mixed water (salinity of 25–31 and sediment < 100 mg/L). The Yangtze River estuary plume is differentiated by the isohaline 25 between the two waters. The Yangtze estuary plume area is a high-quality ecological environment that promotes the growth of marine organisms. This is why the influence of dam construction on the Yangtze River (especially the Three Gorges Dam construction) on the Yangtze plume has gradually become the focus of much attention.

The land satellite images from 1974 to 2009 illustrate how the sediment flux from the Yangtze River into the sea has significantly decreased, a decrease that is strongly correlated with the construction of the Three Gorges reservoir. Upon completion of the Three Gorges Dam, the structure and distribution of nutrients near the plum changed significantly. Wang et al. analyzed the influence of the Three Gorges Dam on the biogeochemical processes in

the downstream reaches of the Yangtze River and the Yangtze River estuary plume [79]. After the Three Gorges Dam finished seasonal runoff from the Yangtze River, the estuary plume decreased by 12–17% in October and increased by 5–20% in the dry season. In addition, due to the decrease of sediment fluxes, the erosion of underwater deltas and coasts, and altered benthic structure increased. From a positive perspective, the interception effect of the Three Gorges Dam on nutrients could alleviate eutrophication in the Yangtze River estuary due to the decrease of nutrient fluxes.

## 4. Management Strategies

Some scholars have explored how reservoir management can mitigate eutrophication in downstream rivers and adjacent sea areas. For example, targeted control of dam storage can increase P and N retention so as to reduce their downstream transfer [118]. At the same time, the reasonable release of stored water during the growth of diatoms can avoid Si limitation in the diatom communities of estuaries and adjacent sea areas [119]. In addition, the targeted control of sediment flushing operations and removing tributary low-head dams could decrease harm to ecosystems in downstream reaches [39,120]. The banning of fishing activities on the Yangtze River and the removal of unnecessary dams (obsolete or small dams) are means of balancing social benefits and environmental health [121,122]. By exploring new reservoir management strategies and performing ecological environmental planning before dam construction, we may be able to balance the social benefits and environmental consequences of dam construction. Overall, there is significant space for the continued development of dam management in the future.

## 5. Conclusions

Rivers carry water, sediment, and nutrients downstream to the ocean. Multipurpose dams are constructed in order to balance economic benefits and environmental costs. The environmental impacts of the dams should be considered before and after dam projects are undertaken. Most studies focus on the effects of dams on reservoirs and downstream rivers, but a relatively small number of studies focus on estuaries and their adjacent marine areas. For the research object, only 32 papers allowed us to draw some very broad general conclusions about the effects of dams on biodiversity in downstream river areas and their adjacent sea areas. Dams affect river flow, water temperature, sediment flux, erosion of estuary deltas, composition and distribution of nutrients, structure and distribution of phytoplankton communities, and fish in ecosystems of downstream rivers and adjacent sea areas. Most studies about the effect of dams on hydrological regimes of the estuary and adjacent coastal areas found that dams could reduce river flow, change estuary salinity, reduce sediment flux, alter water temperature, and change geomorphic processes. The literature about the effects of dam construction on nutrients of the estuary and adjacent coastal area mainly focused on the flux of carbon, nitrogen, phosphorus, and silicon nutrients into the sea. Most studies concluded that dams could alter the composition and distribution of nutrients in the estuary and adjacent coastal areas. The effects of dam construction on ecosystems included changing estuary salinity, altering populations of phytoplankton and zooplankton, blocking fish migration routes, and influencing habitat suitability for birds. More attention needs to be paid to the relationship between the impact of dams on plankton and eutrophication. Most studies in the research on dam effects focused on the plants in reservoirs, downstream channels, and lakes, but the studies about the effects of dams on plants in coastal waters were insufficient. There were only a few studies about the effects of dam construction on birds in the estuary and adjacent coastal areas.

To study the influence of dam construction on the ecological environment, it is necessary to monitor and analyze the key ecological factors so as to provide reliable data and a theoretical basis for the subsequent assessments of dams. Due to the ubiquity of dams, the management of the water in rivers has become essential to nation-building. Dams and reservoirs require us to apply integrated water management strategies as de-

mands for hydroelectric, irrigation, recreation, domestic water supply, and environmental requirements all compete for a limited supply. In order to balance the social benefits and ecological impacts of dam construction, new reservoir construction and management strategies should be explored. Mitigation policies made by the government are a means of ecological rehabilitation. Under the guidance of sustainable development concepts, continuous improvements in technology and experience will provide scientific support for dam construction that effectively utilizes and protects the ecological environment.

Under the concept of the 'Great Protection of the Yangtze River', an increasing amount of attention has been focused on the ecological impacts of dams on the Yangtze River, especially the Three Gorges Dam. The impacts of the Three Gorges Dam on the ecosystems of the Yangtze River estuary can already be observed, but larger long-term impacts on the ecosystem may remain hidden. Therefore, a more comprehensive, systematic, and in-depth study is needed to assess the impacts of the Three Gorges Dam on the marine ecosystem, especially regarding the Yangtze plume.

The impacts of dam construction are not limited to its direct effects on downstream hydromorphology, ecosystems, and human life, but extend to the ecological environments of the estuary and its adjacent marine area. Balancing social and environmental benefits is an ongoing subject, so we suggest that: (1) more comprehensive and effective dam and reservoir management strategies should be explored in order to reduce the effects of dam construction on ecosystems; (2) effects of dams on fish habitats and aquaculture in upper and downstream reaches need further study; and (3) the effects of dams on plants, fish, birds, and animals in downstream and adjacent marine waters be included in future studies to ensure a comprehensive assessment.

**Author Contributions:** Conceptualization, X.Z. and Y.S.; methodology, X.Z.; software, X.L.; investigation, C.F.; data curation, Y.W.; writing—original draft preparation, X.Z.; writing—review and editing, D.H.; visualization, C.F.; supervision, D.H.; project administration, Y.S.; funding acquisition, D.H. All authors have read and agreed to the published version of the manuscript.

**Funding:** This study was financially supported by the National Key Research and Development Program of China (No. 2018YFC1407604).

**Institutional Review Board Statement:** Not applicable.

**Informed Consent Statement:** Not applicable.

**Data Availability Statement:** Not applicable.

**Conflicts of Interest:** The authors declare no conflict of interest.

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
