# Peer review of "Review of Effects of Dam Construction on the Ecosystems of River Estuary and Nearby Marine Areas"

_sustainability, doi:10.3390/su14105974_

Round 1
Reviewer 1 Report
Please find the attached file for comments to authors.

Author Response
Dear Editor and reviewers:
Thanks very much for your kind work to our manuscript (sustainability-1632121). The manuscript has been revised carefully according to the reviewers’ valuable comments. Please see the attachment.

Reviewer 2 Report
The review paper stresses a lot on Three Gorges dam on Yangtze River. Based on the title more examples are required regarding other dams in China and the world. This review is too narrow and seems like a case study of Three Gorges dam. Also a lot of information is not cited, please cite statements that are not general knowledge.
L 35-37: Please cite the statement
Introduction: Please consider citing more papers in introduction.
L 73: please rephrase the statement “1) continue….topic”
Table 1: Change “Reduces” to “Reduction”
L 132-134: Please cite the statement
L 142: Confusing statement. Please rephrase.
L 149-150: Please cite the statement
L 152: Please rephrase
L 165 – 176: Please add adequate references in the paragraph
Section 3.4 – L 206- 223: This section is not adequately referenced.
Conclusion: The conclusion is not based on reviewing and studying most big dams around the world. The generalization used while concluding is faulty.
Author Response

(The authors gave the same response as above.)

Reviewer 3 Report
The study of the effect of dams on rivers has been analyzed in multiple articles. The authors have made an effort to systematize scientific knowledge. However, the structure of the review is not complete enough. The introduction does not highlight the novelty of the review of existing scientific knowledge.
The methodology is weak. It is limited to a keyword search in ISI Web of Knowledge, without indicating the way to systematize the information. For example, the authors treat river systems without considering their geographical location or hydrological regime, key aspects to consider in the analysis, an aspect that is replicated in the results.
Author Response

(The authors gave the same response as above.)

Reviewer 4 Report
I have read with interest your paper on Review of Effects of Dam Construction on the Ecosystems of River Estuary and Nearby Marine Areas. Presented problem is interesting and dynamically developing in the whole world, however, I have a few primary as well as secondary comments.
First of all, the order of the ‘Results’ chapter is logically inacceptable. The analysis is divided into categories: ‘sediment flux’, ‘nutrient flux’, ‘ecosystems’ and ... ‘Yangtze River’ ?!. It makes analysis totally incomparable, leading to crash any conclusion. Moreover, the title of the paper absolutely does not match the content. There are two advises for the authors:
a) change the title of the manuscript, stress the impact of the Yangtze River dams on the estuary in comparison to other rivers (basins)
or
b) change the structure and improve seriously the results chapter.
Apart from the above decision, the ‘results’ chapter needs to be expanded in terms of number and quality of processes as well as effects structure. Not only ALL of diagram components (Fig. 2) should be analysed in details, but many other missed by the authors – e.g. changes in flow variability, changes in estuary salinity after flow recession, results in fisheries etc. At the present state text is too synthetic. Moreover, its part concerning Yangtze River lacks the scientific problems.
The conclusion chapter is hardly connected with main body. Moreover, in review article valorisation of determinants, factors, processes are expected. It should be explained which determinants are more important than the others or there is stochastic pattern of determinants depending on region or something else.
To sum up, the manuscript should be widely revised.
In the first part of text, I have found some secondary mistakes:
- Fig. 1 is insufficiently described.
- line 85: ‘52’ is much more than ‘a few’
- Fig. 2 - regulating river flow by a dam has many more faces than flow reduction only. Give other examples.
Author Response

(The authors gave the same response as above.)

Round 2
Reviewer 1 Report
Overall, I applaud the authors and the extent to which they accommodated my suggestions.
Reviewer 2 Report
Thank you for incorporating the suggestions to the manuscript.
Reviewer 4 Report
good